# Cardiovascular disease prevention: Community Based Asset Mapping within religious networks in a rural Sub-Saharan African neighbourhood

**Andrew Willis[1,2]***, **Samuel Chatio[3], Natalie Darko[4,5], Engelbert A. Nonterah[3,6], Sawudatu Zakariah-Akoto[7], Joseph Alale[3], Ceri R. Jones[8], Ffion Curtis[9], Setor Kunutsor[5], Patrick O. Ansah[3], Sam Seidu[5]**

**1** School of Public Health, University College Cork, Cork, Ireland, **2** HRB Clinical Research Facility, University College 8 Cork, Cork, Ireland, **3** Navrongo Health Research Centre, Ghana Health Service, Navrongo, Ghana, **4** NIHR Leicester Biomedical Research Centre, University of Leicester, Leicester, United Kingdom, **5** Leicester Diabetes Centre, University of Leicester, Leicester, United Kingdom, **6** Julius Global Health, Julius Centre for Health Sciences and Primary Care, University Medical Centre Utrecht, Utrecht University, Utrecht, the Netherlands, **7** Noguchi Memorial Institute for Medical Research, University of Ghana, Accra, Ghana, **8** Department of Psychology and Vison Sciences, University of Leicester, Leicester, United Kingdom, **9** Institute of Population Health, University of Liverpool, Liverpool, United Kingdom

* awillis@ucc.ie

**Data Availability Statement:** All relevant data are within the manuscript and its Supporting Information files.

## Abstract

Prevalence of conditions which raise cardiovascular risk, such as hypertension and type 2 diabetes are seeing a dramatic rise in Sub Saharan Africa. A large proportion of these cases remain undiagnosed and there is limited resource to provide patients with self-management support and education once diagnosed. This study aimed to identify and catalogue community-based assets for the purposes of developing and deploying a screening and education programme for cardiometabolic risk factors (diabetes and hypertension) within religious organisations in a local community in a rural Ghanaian context. We utilised a community-based form of participatory research made up of a number of different components including community-based asset mapping and stakeholder consultation, supplemented by 18 in-depth interviews and 10 focus groups with n = 115 service users, to map existing assets with relevance to cardiometabolic health in this setting and context. Thematic analysis of interview and focus group data was performed to identify themes related to successful implementation of health screening. Two stakeholder workshops with local healthcare professionals, faith leaders and health policy makers were delivered to co-produced a prioritised list of recommendations and 'asset map' to aid deployment of mass screening within faith organisations in this context. The findings of this research highlight a number of 'hidden' community assets and motivational mechanisms at an individual, community and institutional levels; these have informed a list of recommendations which have been co-developed with the stakeholder group and local community to support the development of effective screening strategies for cardiometabolic conditions within faith organisations in this context. We have identified key mechanisms and assets which would support a sustainable

**Funding:** SS, AW, ND, CRJ and PA were in receipt of funding from European Foundation for the Study of Diabetes and Lilly through the 'Exploring and applying new strategies in diabetes (EXPAND) programme. The funders had no role in study design, data collection and analysis, decision to publish, or preparation of the manuscript.

**Competing interests:** I have read the journal's policy and the authors of this manuscript have the following competing interests: SS is in receipt of: Speaker honoraria from: AstraZeneca, Boehringer Ingelheim, Janssen, Lilly, MSD, Novo Nordisk, SB Communications, OmniaMed, Roche, Napp, NB Medical, Amgen, Advisory board honoraria from: AstraZeneca, Lilly, Boehringer Ingelheim, Janssen, MSD, Novo Nordisk, Takeda, Sanofi, Educational grants from: Boehringer Ingelheim, Lilly, Novo Nordisk, Takeda, Conference registration and subsistence from: Boehringer Ingelheim, Janssen, Lilly, Novo Nordisk, Takeda. All other authors have declared that no competing interests exist.

screening approach designed to engage an underserved community at high CVD risk to promote general community health and well-being.

## Background

The burden of non-communicable diseases (NCDs) is growing at an alarming rate in Sub Saharan Africa (SSA) [1], with recent prevalence estimates ranging between 3.0–13.9% [2] and 15.1–54.1% for diabetes and hypertension, respectively [3]. It is of particular concern that for both conditions, most cases remain undiagnosed [1] with missed opportunities for intervention to reduce CVD risk. Projections from the International Diabetes Federation show that whilst globally, diabetes prevalence is expected to increase, the largest increase will be seen in SSA [4].

The sector commonly known as the faith-based non-profit (FBNP) sector in Ghana has predominantly evolved from Missionary organizations during the colonial era. It encompasses providers from various religious backgrounds, including religion-oriented non-governmental organizations, informal community initiatives, and the Ahmadiyya Muslim Mission (AMM) [5].

There is a large body of previous literature which acknowledges religion as an explanatory variable in health outcomes [6]. Religion, and one's religious beliefs can be seen to encompass behavioural, attitudinal, public, and private activities, all of which have the potential to impact upon health-related behaviours and health outcomes [7]. Increasingly, researchers and public health professionals have sought to capitalise on the positive influence of religion and its shaping of health behaviours, using faith-based, and faith-placed health intervention programmes to improve access to healthcare and disseminate health education [8].

Particularly in SSA, literature on the use of faith-placed health intervention is relatively sparse; the little literature that exists focusses predominantly on communicable disease prevention/management, particularly HIV [9]. Given the increase in the burden of NCDs such as type 2 diabetes (T2D), there is an urgent need to establish simple, low-cost scalable interventions which harness and mobilise existing assets and in other settings, to address this sizable public health challenge.

Community-Based Asset Mapping (CBAM) has extensively been used in the field of community development and provides a strength-based approach to focus on what assets communities already possess [10]. The CBAM approach views a community as a place with strengths or assets that need to be preserved and enhanced, not deficits to be remedied. These assets can be reconfigured to leverage increased value in achieving a specific goal [11]. There are a number of frameworks that exist which describe methods of asset mapping in both religious and non-religious contexts. There has been limited application of this approach within a healthcare context; in low- and middle-income countries (LMICs) such as those in SSA, the use of this approach has focussed on the management of communicable diseases [12].

In the current paper, we describe the process of mapping out assets at an individual, community and institution level in a setting in rural Ghana, which exist in and around religious communities and congregations and which have relevance to physical and mental health, with a particular focus on CVD risk. These findings are presented, together with recommendations relevant to the development of faith-based screening programmes for CVD risk factors.

### Self Determination Theory

Self-Determination Theory (SDT) is useful theoretical framework to understand individual's motivational mechanisms to engage in positive health behaviours and health screening. SDT is

a theory of intrinsic motivation [13] which has been applied to a number of different contexts including education [14], health behaviour change [15], and pro-environmental behaviour [16]. SDT assumes all individuals have the same three innate psychological needs, competence, relatedness and autonomy. The need for competence is deemed to be satisfied through successful completion of a goal or mastery of a skill alongside the ability to effect change [13]. The need for relatedness refers to a desire for acceptance from others and is deemed to be satisfied through the development of social connections and interpersonal relationships [13]. Finally, the need for autonomy is deemed satisfied when an individual feels able to make their own decisions without restriction or pressure from others. SDT research has demonstrated that these three motivational constructs are consistent across cultures [17–19].

Psychological theories such as SDT are useful to understand the intrinsic motivational mechanisms that underpin engagement and uptake of CVD (e.g. T2DM and hypertension) screening to in rural Northern Ghana. CBAM can identify the social environments and assets that may support or hinder motivation to engage. Recent research has sought to combine both SDT and CBAM [20] to strengthen the implementation and evaluation of community based interventions.

## Methods

### Ethics statement

Ethical approval was granted by the Navrongo Health Research Centre Institutional review board ref: NHRCIRB47I on 27/07/22. Prior to participation, informed consent was collected from all participants. Study staff provided translation services for those unable to read English, written consent was obtained where possible, with fingerprint consent taken from participants who could not sign.

### Setting

This work was carried out in the Kasena-Nankana East Municipality and Kasena-Nankana West District in the Upper East Region of Northern Ghana. The two districts fall within the coverage area of the Navrongo Health and Demographic Surveillance System (HDSS) operated by the Navrongo Health Research centre (NHRC). The area has previously been found to be particularly underserved taking into account rurality and distribution of current health assets provided by the national health system [5].

### Study design

We used a method of Community Based Asset Mapping (CBAM) based on the previously published Participatory Inquiry into Religious Health Assets, Networks and Agency (PIRHANA) model [21]. All CBAM approaches are participatory approaches [22] to development that are internally focussed on and help community members themselves identify capacity to problem solve with an emphasis on community resources rather than external resources. The approach is relationship driven and can build the social capacity of participating community members, establish and nurture ongoing relationships between community members [23]. The PIRHANA model was used to inform our approach as it has been specifically designed for use with faith communities and used extensively in SSA. Although the approach chosen was the most appropriate available, by nature, most models for asset mapping are highly context and condition specific [23] and further work will be required to develop and adapt screening models in order to optimise this approach in this setting.

## Target participants and sampling

In October 2022, the research team at NHRC worked collaboratively with local churches and religious organisations, faith leaders and traditional and faith healers, healthcare professionals, public health officials and community members who seek health services within the communities in the Kassena-Nankana East Municipality and Kassena-Nankana West District in the Upper East Region, Ghana.

Purposive sampling [24] was carried out by the research team, in partnership with local faith and community leaders who identified participants based on their religion, membership of the relevant mosque or church congregation, age and gender to ensure that community members were suitably diverse in terms of gender, age, religion (Christianity, Muslim etc,) and socioeconomic status, and the views and insights were reflective of the target population who would have access to faith-based CVD risk screening services. Participants were identified and approached in person via community leaders who, together with the researchers, explained the purpose of the study and what their involvement would entail. Consented participants were provided with a mutually convenient appointment for a Focus Group Discussion (FGDs) or of in-depth-interview (IDIs).

FGDs, IDIs and workshops were convened at a central research facility and within church or mosque and health compounds with participants given non-financial-incentives (refreshments and hygiene products) or financial reimbursement for travel time and this was according to recommendations of the ethics committee.

## Data collection

The research team hosted a series of workshops, focus group discussions and in-depth interviews which were inclusive and catered for individual needs including; language translation, running female only FGDs and holding groups in rural locations to address issues of access. FGDs and IDIs were based on an initial semi-structured topic guide which sought insights on the following: the context of religion and cardiovascular health in this community, the key factors which promote or harm health, identification of organisations surrounding churches which influence health in the locality, how the church/religious organisations can contribute to CVD/T2D treatment care and prevention and what next steps can be taken as a result of the data collected. Focus group discussions (FGDs)and in-depth-interviews (IDIs) were within church or mosque and health compounds. These settings were chosen due to their close proximity to the communities of interest, the spaces used were familiar to participants which aided more in-depth discussions. The choice was based on recommendations from local co-investigators who have experience and good ongoing relationships with local communities from previous research studies. Workshops were carried out at a central research facility attached to the local hospital. The majority of FGDs and IDIs were facilitated by three junior research staff (research assistants), who lived within local communities in Navrongo and were members of either the Christian or Catholic Church and are fluent in local languages (Kasem and Nankani). Workshops were facilitated by four members of the research team from Ghana and the UK who were experienced in qualitative research and all had a background in social sciences and public health. An iterative consultation and co-design process was used during the workshops, FGDs and IDIs. Researchers produced resources and graphics based on initial discussions which were refined following each discussion to produce final versions of the resources. All workshops, FGDs and IDIs were recorded and translated by bilingual research assistants and transcribed verbatim.

## Data management and analysis

Data analysis was composite, whereby we conducted thematic analysis [25] of community members, Healthcare Professionals (HCPs) and policy maker's insights relating to the

mapping of health services provided in the locality. Geographical maps of local religious organisations and churches (>90) was compiled with contact details for faith leaders and wider organisational members. Graphical illustrations of community assets were co-developed with participants together with a list of recommendations of 'wish list' for the faith-based CVD screening programme, which is currently being developed as part of the wider study. Transcripts were coded by two researchers independently using NVivo 17. The thematic framework developed (S1 Fig) and an example of the coded interview transcripts (S1 Data) are provided as a supporting information files.

Salient themes and specific findings which informed the screening approach and design are described below. Key findings and all resources were shared with community members and faith leaders during a second round of interviews to promote accurate interpretation of data and confirmability of findings. Recommendations for screening were ranked and presented according to perceived importance from both a healthcare user (congregation member) and provider (HCP or faith leader) view.

## Results

Two stakeholder workshop discussions were held at a central location—NHRC conference hall and attended by 21 participants including nurses, faith leaders, public health professionals, traditional healers and medical doctors. Staff conducted an additional ten FGDs (5 male 5 female) in local churches, mosques and community settings. A total of 115 people attended the ten focus groups, attendees were all community members regularly seeking healthcare. A further 18 IDIs were conducted in local faith and community settings, interviewees were faith leaders (n = 6), faith nurses (n = 6) church elders (n = 3), women's group leaders (n = 1) and (youth group leaders (n = 2). Full details are provided in Table 1. Community settings which hosted the screening included mosques and both catholic and other Christian churches. These settings are referred to in the results below as 'faith centres' unless explicitly referred to when relevant to one's religious community.

'*Assets*' were grouped into three themes, at the individual, community and institutional level (Fig 1). Notable in the discussion around individual assets were the existing clinical knowledge and skills held by congregation members who are trained healthcare professionals. The trust in their experiences and decision making regarding healthy lifestyle, including attitudes towards traditional medicine was evident. Community members and nurses spoke about tensions between healthcare professionals and traditional medicine and faith healers regarding the efficacy of some of the treatments used.

There was much discussion with individual faith leaders regarding their specific knowledge of scripture, and particular bible passages which could be included in church sermons to encourage an individual to engage in preventative healthcare and increase willingness to engage in screening for CVD risk factors. Also evident was the position of the faith centre as a social 'hub' within the community giving a sense of 'belonging'. Altruism and volunteering from congregation members were critical to many of the wider functions of the faith centres. This sense of responsibility extended beyond just their own congregation with a feeling that 'Jesus came for everyone, not just for those from the church', the implication being that the whole community should be invited to health promotion activities, not just those attached to that particular faith centre.

Community level assets that were cited during discussions included the various management and welfare groups which are involved in the day to day running of each faith centre. Specified roles of these groups included management, advocating for social action or providing financial and other support in times or hardship which could potentially support a small number of people who screen positive for CVD risk factors and have no way of accessing

**Table 1. Participant socio-demographic characteristics.**

| Total n = 115 | Focus group Participants n = (%) |
|---|---|
| **Gender** | |
| Male | 56(49) |
| Female | 59(51) |
| **Age in years** | |
| 20–30 | 21(18) |
| 31–40 | 31(27) |
| 41–50 | 36(32) |
| 51+ | 27(23) |
| **Religion** | |
| Christian | 90(78) |
| Muslim | 25(22) |
| **Educational status** | |
| Never went to school | 22(19) |
| Primary school | 31(27) |
| Secondary school | 37(32) |
| Tertiary/Professional | 25(22) |
| **Employment status** | |
| Unemployed | 7(6) |
| Civil/Public sector | 22(19) |
| Farmer/Artisan | 27(23) |
| Trader | 41(36) |
| other | 18(16) |
| **Ethnicity** | |
| Kasem | 94(82) |
| Nankani | 3(3) |
| Other | 18(16) |

medication. Elders' groups, women's groups and youth groups helped to engage congregation members across different genders and age ranges. Community members and faith leaders also spoke about particular cultural and religious events which could provide opportunities for mass screening. Participants also described traditional leadership structures including town *'Chiefs'*, *'Tindana'* (referring to land owners) and *'Nakwa'* (referring to elders) and the potential ways that people in these positions could influence health behaviour locally through leadership, [26] advice and activism.

## Stakeholder recommendations on intervention

Focus group and workshop participants spoke about integration of the wider health service and involvement of staff as assets to leverage additional skill, legitimacy and trust including pharmacists, chemists, CHPS compound staff [27], hospital staff and staff from regional health facilities. Participants also discussed stakeholders involved in the publication of health messaging including local radio stations, local town opinion leaders or 'chiefs' and the role of social media in engaging younger generations. Owing to the security situation in the local area, participants identified local police and security services as assets which could improve trust and likelihood of attending large events if security can be assure

Thematic analysis of views expressed in both the FGDs and IDIs yielded a number of common themes, sub themes and recommendations (Table 2) necessary to facilitate a 'successful'

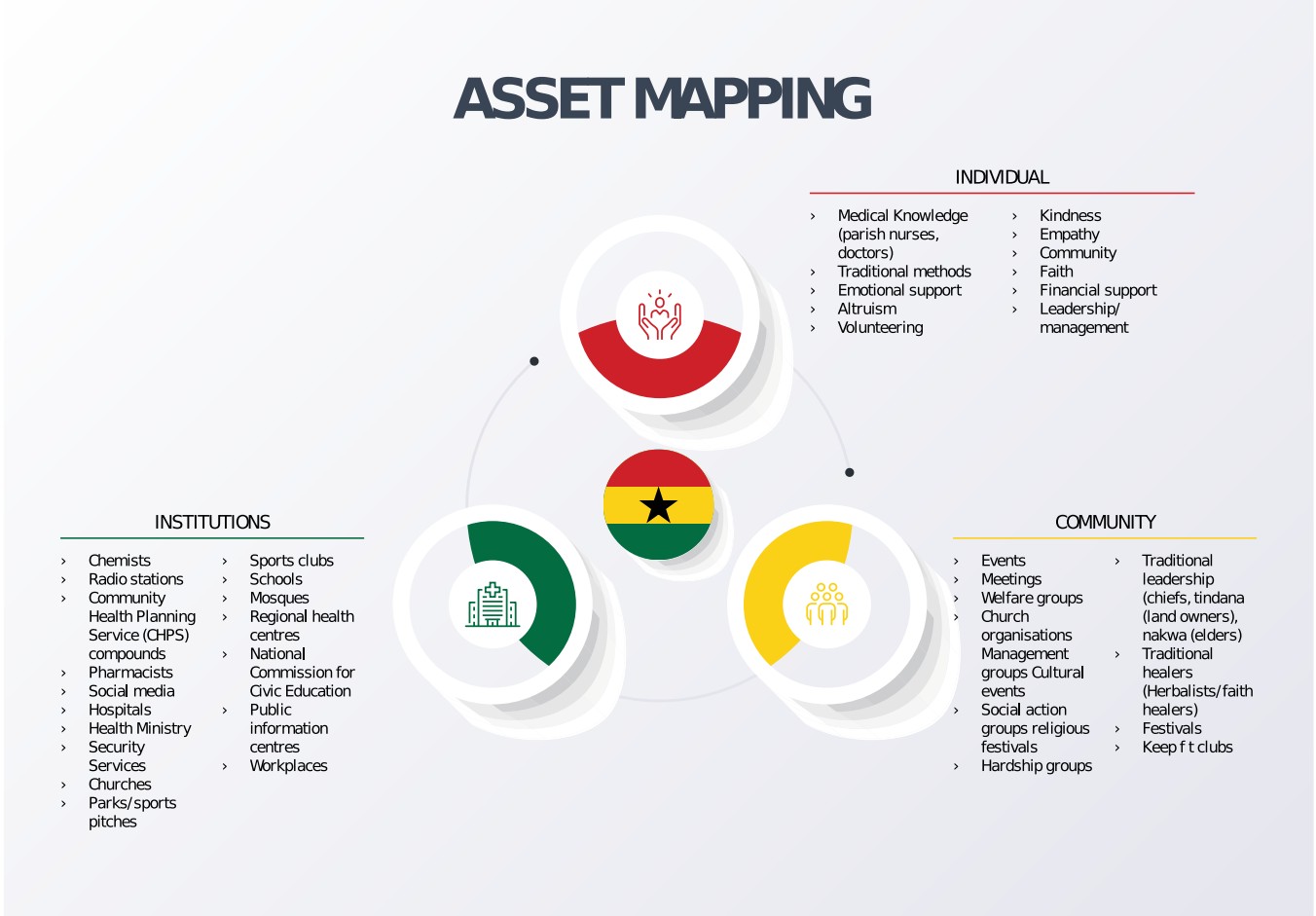

**Fig 1. Asset map.**

screening programme characterised by high uptake and sustained engagement. These recommendations have been co-produced by congregation members, faith leaders and health professional and are presented in order of priority (Table 3).

Participants prioritised operational factors including timing and location as key facilitators for success. In the rural location in which the study took place, a large proportion of the population are employed in the agricultural industry and it was felt that avoiding busy work periods during harvest (October-January) would make people more likely to attend mass screening. Members of the Muslim community felt that timings chosen would also need to respect prayer times during the day.

Congregation members were clear that confidentiality would need to be ensured during appointments as people are uneasy with disclosing disease status even within their own community. Steps would be needed to make sure that feedback and advice after screening would be on an individual basis to prioritise confidentiality.

Previous negative experiences of interactions with health services and screening events manifested in a level of mistrust in congregations. Participants felt that it would be important that the screening team were clear on their goals and open about motivations for providing screening. This 'meeting of expectations' from those undergoing screening, regards to number

**Table 2. Asset mapping responses.**

| Asset classification | Illustrative Quotes |
|---|---|
| *Individual level* | Teaching/Leadership *"I think almost all of us here are teachers and so, in our work places and schools, we can educate people. I have always said that I am not a health worker but I am health conscious" (P6, female 18-40yrs)*<br>Competence: *(medical knowledge)"Fortunately, we have members that are nurses in our church, so if you inform them, they can help." (P8, female 18–40 years)*<br>*"We have those who preach, so all those people can be the counsellors" (P12 Male, 40-60yrs)*<br>Leadership; *"I can remember when they were doing the vaccination for COVID-19, people were having this perception that if you take the vaccine, you will become impotent or barren. So, they started with the health workers and we those that are working with them. So, we those who took it first then communicate to them that all those rumours are not true, and now they are even running to take the vaccine" (P4 Male, 40+yrs)*<br>Traditional Knowledge: *"In the olden days, kassena's had a lot of medicines that can help. It is recently that we abandoned them. 'Tontogo' (grass) added with dawadawa and red gravels (solosogo) can help cure diabetes. You soak them in water together and sieve the water and be drinking". (P3, Male 40+ years)* |
| *Community level* | Traditional leadership: *"To me, if the whole process is to begin, you should inform the chiefs and elders of the community because they have knowledge of the time the associations within the community"* (P9, Female, 40+yrs)<br>Hardship funds *"The church can have a fund like that to help people with certain conditions. If there are funds and people come with such conditions, the church can pick money from the fund and see to it that they are given medical attention" (P6, female, 40+)*<br>'Relatedness': Church groups: *"The men's group for instance, meet together to socialize and also find means and ways to reach their fellow elderly people and the same applies to the youth too". (P2, Male 40+)*<br>Cultural events: *"We are aware the churches and perhaps the mosques also have some annual conferences which usually they have health screening as part of it. So, if we are able to identify these on the calendar, perhaps you can ride on that so that the small that you are putting helps". (P19, Male 40+yrs)* |
| *Institutional level* | Faith Centres *"The church and the other bodies should help to absorb some of the cost of the drugs. This will help those who have diabetes to get full treatment. The churches too can also make offerings or collections to raise funds to support the treatment of such diseases"* (P8, female 18-40yrs)<br>Security Services: *What I want to suggest is that such screening is going to involve a lot of people so if we can involve the security. Where we find ourselves is close to the border and lots of things normally go on so that no one should use that as an advantage. (P1, male 40+yrs)*<br>Autonomy: Faith Healers: *"We have really done a lot but I think on the part of the community, we can look at the healing centres at the community level. The reason why I am saying the healing centres is that a lot of community members trust in those people and even sometimes before they even go to the hospitals, they go there first"(P4 female, 40+yrs)* |

of tests available, how long a test would take, data security and availability of follow up support was vital to participants.

This 'follow up support' was seen as highly important in the context of a clear plan for maintenance of the programme and providing some form of service for people who screen positive or are identified as being at increased risk. Congregation members and faith leaders showed a need for clarity on who would fund medication which would be required and if there was an expectation for people to pay themselves. A small number of congregation members and faith leaders spoke about the viability of setting up welfare groups within their own organisation which would have responsibility for educating people on matters relating to health, as a low-cost way of supporting their congregation on an ongoing basis, utilising existing assets.

## Discussion

The scale of influence of religion and religious institutions at an individual, community and institutional level in Ghana and across SSA is widely documented. If health interventions are

**Table 3. Stakeholder recommendations.**

| Recommendation | Illustrative quotes |
|---|---|
| *Timing & location* | *"Normally after mass on Sundays, some people are always time conscious and want to run away. So, if you want to keep them any longer, you may have issues with them"* (P8, female 18-40years)<br>*"A day like market days, would not be good for most people because they go to the market and when you want to delay them, they will have issues with you"* (P2, female 18-40years)<br>'*After harvesting. Now people are harvesting till November. So, after November going to December and January will be good. So, basically the dry season will be good for the screening.*' P9 Male 40+) |
| *Publicity/Communications* | *"The social media also helps a lot especially the radio stations. If there are outbreaks of certain diseases, they announce and educate people about those conditions so that they know what to do to overcome such conditions"* (P1, female, 18-40yrs)<br>*"I think fliers and posters can be designed where you put the causes, effects of it in picture forms. When people see it, they will know that when I get it, this is what happens to me. That can be a motivation to them to avail themselves for the screening exercise and for further health care"* (P6, female, 18-40yrs) |
| *Importance of pre-screening education* | *"If we are educated on the causes and the prevention, it would open up our minds to know those ways of life that predisposes us to those conditions."* (P8, female, 40+yrs)<br>*"The person has not seen the effects of the disease that much so, education about the disease will help to a greater extent. This will make the person get to know what the disease is capable of doing to him and so the need to get treatment as early as possible."* (P1, female, 18-40yrs) |
| *Meeting expectations* | *"When you promise to come next year, please come next year. The foundation has been laid so let us build on it up to the top. But if you disappoint the people, next time, the organization will be difficult for the leaders".* (p3, male, 40+ years)<br>*"If you say you are using 30 minutes to do the screening, the people will program their stomach knowing that he/she is going to sacrifice 30 minutes without food, so if you abuse those 30 minutes without satisfying his/her stomach, they may leave in the middle of the activity"* (p3, male, 40+ years) |
| *Confidentiality* | *"You cannot even disclose someone status in front of other people because it doesn't sound good. People cannot control their mouths so they will spread it fast. So individually will be best (P2, female, 40+ years)*<br>*These diseases are not shameful ones if you are tested positive. It wouldn't be a big deal but you know we humans talk a lot, so it will be better to disclose the results individually than publicly."* (P4, female, 40+ years) |
| *Clear referral pathways* | *"If it is for you to come and test and not give drugs at the end, I will not test because I have no money to be able to afford the drugs. If you know you will give us medicines after testing, then we will test. You can see that times are hard."* (P11, female, 40+ years) |
| *Plan for maintenance* | *"I think bringing health personnel outside the church will be nice. The health personnel in the church will serve as backup and work together and the health workers in the church should be trained so that after you left, then they will take over"* (P3, male, 40+ years) |
| *Support from faith leaders* | *"You know that they have leaders in every organisation. They have elders in the church and here too we have elders. Not anyone can advise someone to understand. Take me for example, if I am to go with this lady, it will have weight because if you are describing it, she can also describe it in a different and clearer way"* (P1, male 40+)<br>*"If you go to most of the houses and introduce yourself as a worker of VAS, they will not take you seriously. They will say they do not see the good in what you are going to do but if you pass through the mosque and I also go home to spread the message, they will believe and they will come and take part."* (P6, male 40+) |
| *Post-screening counselling* | *"I think educating such a person will work. At a point some people lose hope because they don't have anyone to help them so they might not go when you refer them. So, I think if you educate the person very well, he/she will be convinced and also people like you can also help in one way or the other."* (P7 female 40+)<br>*"Counselling is good and will help all of us. Because if you test positive then the counselling will let you know how to get rid of it and if you test negative, you can protect yourself against it."* (P1 female 40+) |

(*Continued*)

**Table 3.** (Continued)

| Recommendation | Illustrative quotes |
| --- | --- |
| *Co-Development/delivery* | *"As you said the pastor can include the health education in the preaching to create awareness about the activity. that is one way because the pastor is in this church and he/she is helping us to create awareness"* (P1, Male, 40+) |

to be successful in this context, clearly, they require extensive collaboration and co-delivery, together with a deep understanding of the networks and assets that exist in and around faith centres in order to address the reported challenges of implementation within this context inclusion difficulties with evaluation [28], paperwork and conflicts in views and beliefs between modern medicine and traditional methods of healing [29].

This study used asset mapping to promote better understanding of these networks and identify an extensive list of existing assets, (skills, knowledge, community assets & institutions) which influence health behaviours relevant to CVD prevention. Furthermore, a diverse group of congregation members, healthcare professionals and faith leaders shared their views on how these assets could be best utilised or reconfigured to successfully deliver health screening within faith centre settings in the study area.

The study uncovered social structures and organisations within faith centres which could provide opportunities for maintenance and ongoing support, institutions and wider stakeholders including the police and security services who would be needed to provide security at large gatherings. Stakeholder discussion allowed exploration of the relationships between assets, including the complex relationship between traditional (faith healers, traditional healers) and modern medicine (nurses, pharmacists, doctors). Workshops were used to produce a set of recommendations for operationalising screening which brought in a number of the assets that were identified, notable existing clinical knowledge/skills from congregation members, championing from faith leaders, including supporting messages from scripture in sermons which may encourage self-care and screening-uptake and the possibility of follow up care through existing social welfare groups.

SDT can be used to understand the motivational mechanisms of engagement supported by local assets identified through the CBAM. The church and its members support important motivational mechanisms for engagement alongside the existing assets within the community. Harnessing assets such as existing knowledge and skills within the church via faith nurses, and those who preach could educate and develop individual knowledge and competence. Individual community member's competence and understanding of the importance of CVD screening and actively managing CVD risk factors can be achieved through the church and their preachers working alongside healthcare professionals to share the knowledge and encourage participation. Through this education, a sense of mastery can be achieved where an individual feels knowledgeable and capable in managing their own condition. Empirical studies on CVD have demonstrated that where individuals perceive they have limited control over life circumstances this is associated with an increased risk of CVD mortality [30] and conversely a sense of mastery is associated with lower CVD risk index scores [31].

When individuals have the autonomy to decide when and how they engage in screening services then local assets can provide a powerful supportive mechanism. Individuals may choose to engage with healing centres, which serves to align the intervention with the more traditional African healing paradigm [32]. Thus, autonomy is created when individuals have the agency to act and engage in way that suits their individual needs. Results suggest that existing groups (men's, women's, youth and elderly) can be utilised to support relatedness and model the importance of screening participation. Relatedness is created when peers and

important individuals in the church are seen to be engaging. These social connections can be further bolstered by engagement in CVD screening creating a sense of acceptance in those peer groups. Previous research has demonstrated that peer social influence is a powerful mechanism for overcoming medical mistrust [33].

Several findings of this research are similar to what has been reported globally in identifying facilitators for faith-based health promotion. Specifically the role of faith leaders in influencing health behaviour of congregation members [34], the use of existing social welfare clubs focussing on chronic disease management and education [35] and church attendance and religion as a means of increasing social capital [36]. The findings of this study also provide further evidence showing priorities integral to the design of future health interventions, particularly around disease detection and mass screening services.

## Implications of the findings

The findings of this study have important implications for both research and clinical practice within this context. The asset mapping combined with SDT provide a framework to understand motivations to engage and highlight opportunities to build in maintenance and sustainability, at low cost, utilising existing assets and skills which already exist within religious organisations, including medical knowledge and social welfare clubs. This could provide a means of overcoming one of the most significant challenges for implementation of embedding the screening within a health system with limited resource, particularly focussed on primary prevention, at low cost.

## Strengths & weaknesses

This study utilised a novel approach to identifying assets and stakeholders involved in provision of health screening in a rural setting in SSA. By using a purposive sampling strategy [37], particularly sensitive to the differing levels of socioeconomic status, religion, gender education level, we have taken into account a wide variety of views to capture this diversity, essential in developing a screening method which takes account of individual barriers and strives to support those who are most vulnerable and suffer a disproportionate disease burden and has provided views through the dual lens of seekers and providers.

Despite the novelty of the methods used, there are limitations associated with focussing on the church. Though the church is an important institution and context in SSA, a proportion of the population may not engage and therefore may be missed via the method of recruitment we have chosen. The findings are likely to be very context specific and their generalisability to outside of Ghana is untested, even to more populous settings such as major cities; however, the work is important as deprived rural communities have lower access to healthcare generally, higher levels of poverty and delayed diagnosis of CVD and T2D [5]. There is a rich variety of religious expression, as evidenced by the diversity of participants and religious entities included here and by the underlying influence of African traditional beliefs and healing practices. There is no 'one size fits all' approach and public health interventions should take account of community assets and intrinsic motivations to engage before designing and delivering these interventions.

## Conclusion

Our findings highlight the diverse skills and resources which directly relate to prevention of CVD and T2D in religious settings in rural Ghana. Furthermore, views from healthcare providers and users provide key recommendations for operationalising health screening in this setting. Further work is required to upskill lay community members with the skills and

knowledge to leverage maximum impact from these assets and to determine long term sustainability of interventions delivered religious settings in this context.

## Supporting information

**S1 Fig. Coding framework.**
(EPS)

**S1 Data. Example theme coded data.**
(DOCX)

**S1 Text. Reflexivity statement.**
(DOCX)

## Acknowledgments

This research was supported by the National Institute for Health Research (NIHR) Collaborations for Leadership in Applied Health Research and Care (CLAHRC)–East Midlands, NIHR Leicester Biomedical Research Centre, and the Centre for Ethnic Health Research. We are grateful for the efforts of Research Assistants David B. Ajebakwagane and Rexford A. Aduah for their efforts facilitating data collection.

## Author Contributions

**Conceptualization:** Andrew Willis, Samuel Chatio, Natalie Darko, Engelbert A. Nonterah, Sawudatu Zakariah-Akoto, Ceri R. Jones, Ffion Curtis, Setor Kunutsor, Patrick O. Ansah, Sam Seidu.

**Data curation:** Samuel Chatio, Natalie Darko, Engelbert A. Nonterah, Sawudatu Zakariah-Akoto, Ceri R. Jones, Ffion Curtis, Setor Kunutsor, Patrick O. Ansah, Sam Seidu.

**Formal analysis:** Andrew Willis, Samuel Chatio, Natalie Darko, Engelbert A. Nonterah, Sawudatu Zakariah-Akoto, Ceri R. Jones, Ffion Curtis, Patrick O. Ansah, Sam Seidu.

**Funding acquisition:** Andrew Willis, Samuel Chatio, Natalie Darko, Engelbert A. Nonterah, Sawudatu Zakariah-Akoto, Ceri R. Jones, Ffion Curtis, Setor Kunutsor, Patrick O. Ansah, Sam Seidu.

**Investigation:** Andrew Willis, Samuel Chatio, Natalie Darko, Engelbert A. Nonterah, Sawudatu Zakariah-Akoto, Joseph Alale, Ffion Curtis, Setor Kunutsor, Sam Seidu.

**Methodology:** Andrew Willis, Samuel Chatio, Natalie Darko, Engelbert A. Nonterah, Sawudatu Zakariah-Akoto, Sam Seidu.

**Project administration:** Andrew Willis, Samuel Chatio, Natalie Darko, Engelbert A. Nonterah, Sawudatu Zakariah-Akoto, Joseph Alale, Ceri R. Jones, Ffion Curtis, Sam Seidu.

**Writing – original draft:** Andrew Willis, Ceri R. Jones.

**Writing – review & editing:** Andrew Willis, Samuel Chatio, Natalie Darko, Engelbert A. Nonterah, Sawudatu Zakariah-Akoto, Joseph Alale, Ceri R. Jones, Ffion Curtis, Setor Kunutsor, Patrick O. Ansah, Sam Seidu.

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
