## [Decision Letter · Decision Letter 0]

11 May 2023

PGPH-D-23-00401

Cardiovascular disease prevention: Community Based Asset Mapping within religious networks in a rural Sub-Saharan African neighbourhood

Dear Dr. Willis,

Thank you for submitting your manuscript to PLOS Global Public Health. After careful consideration, we feel that it has merit but does not fully meet PLOS Global Public Health’s publication criteria as it currently stands. Therefore, we invite you to submit a revised version of the manuscript that addresses the points raised during the review process.

This an important paper. We have received some comments from an external peer reviewer. I have also included my own comments. Kindly go through them carefully and revise your manuscript. 

We look forward to receiving your revised manuscript.

Kind regards,

Vijayaprasad Gopichandran

Academic Editor

Journal Requirements:

2. Please send a completed 'Competing Interests' statement, including any COIs declared by your co-authors. If you have no competing interests to declare, please state "The authors have declared that no competing interests exist". Otherwise please declare all competing interests beginning with the statement "I have read the journal's policy and the authors of this manuscript have the following competing interests:"

3. Please amend your detailed Financial Disclosure statement. This is published with the article. It must therefore be completed in full sentences and contain the exact wording you wish to be published.

Additional Editor Comments (if provided):

Thank you for submitting your manuscript for consideration. We appreciate the efforts at conducting a research on an important topic. engaging with faith based institutions and organisations is an important activity in health promotion especially for non communicable diseases as it involves adoption of new and healthy life styles. However, there are several limitations in the way the manuscript has been written. If these are revised, the manuscript can be made more meaningful and readable. Please note the following comments

1. In the methods section it is not clear how the participants were selected in the study. It is mentioned that it was purposive selection. But the exact details of how they were selected in not provided.

2. Why did the researchers choose to conduct the interviews and group discussions in the faith based organisations like churches and mosques? It would have been better to conduct them at the community level as even people from distances far away from the religious institutions could have participated and they could have brought in the perspective of distance.

3. Who conducted the IDI and FGDs? What is their background? What religion did they practice? Some details of the interviewers would be helpful.

4. A reflexivity note by the interviewers would also be helpful to interpret the findings of the research.

5. More details must be provided on how the data analysis was done. how many researchers did the coding and theme building? what processes were adopted for coding and theorising?

6. The entire paper lacks any verbatim quotes. The authors must provide verbatim quotes along with a brief description of the persons who said those quotes.

7. There are several methods used by researchers to get participants to engage actively in participatory methods. For example, they may be asked to draw maps, they may be asked to sort resources into groups, etc. What participatory action method was used by the authors in this study? This is not explained.

8. There is no mention of what methods were adopted to ensure credibility, dependability, transferability and confirmability of the qualitative research findings. This must be clarified in the paper.

the authors should carefully go through these critical inputs and revise their manuscript accordingly.

Reviewers' comments:

Reviewer's Responses to Questions

**Comments to the Author**

1. Does this manuscript meet PLOS Global Public Health’s publication criteria? Is the manuscript technically sound, and do the data support the conclusions? The manuscript must describe methodologically and ethically rigorous research with conclusions that are appropriately drawn based on the data presented.

Reviewer #1: Yes

2. Has the statistical analysis been performed appropriately and rigorously?

Reviewer #1: N/A

3. Have the authors made all data underlying the findings in their manuscript fully available (please refer to the Data Availability Statement at the start of the manuscript PDF file)?

Reviewer #1: Yes

4. Is the manuscript presented in an intelligible fashion and written in standard English?

Reviewer #1: Yes

5. Review Comments to the Author

Reviewer #1: Comments and Suggestions for Authors

The authors conducted a study to identify and catalogue community-based assets for the purposes of developing and deploying a screening and education programme for cardiometabolic risk factors (diabetes and hypertension) within religious organisations in a local community in a rural Ghanaian context. The paper is very well-written, well-organized, and interesting. They used appropriate participatory approaches and provided an adequate discussion of the results. I have a few minor editorial comments that will hopefully improve the paper.

* The author has mentioned that they have used Community-based Asset Mapping with a different approach. Did the authors consider ways that findings from different stakeholders traditional (faith healers, traditional healers) and modern medicine (nurses, pharmacists, doctors) can be applied in best practice for CVD screening and educational programs?

6. PLOS authors have the option to publish the peer review history of their article (what does this mean?). If published, this will include your full peer review and any attached files.

**Do you want your identity to be public for this peer review?** For information about this choice, including consent withdrawal, please see our Privacy Policy.

Reviewer #1: No

---

## [Editor Report · Decision Letter 1]

1 Sep 2023

Cardiovascular disease prevention: Community Based Asset Mapping within religious networks in a rural Sub-Saharan African neighbourhood

PGPH-D-23-00401R1

Dear Dr Willis,

We are pleased to inform you that your manuscript 'Cardiovascular disease prevention: Community Based Asset Mapping within religious networks in a rural Sub-Saharan African neighbourhood' has been provisionally accepted for publication in PLOS Global Public Health.

Best regards,

Vijayaprasad Gopichandran

Academic Editor